# Safety Culture and the Positive Association of Being a Primary Care Training Practice during COVID-19: The Results of the Multi-Country European PRICOV-19 Study

**DOI:** 10.3390/ijerph191710515

**Published:** 2022-08-24

**Authors:** Bianca Silva, Zlata Ožvačić Adžić, Pierre Vanden Bussche, Esther Van Poel, Bohumil Seifert, Cindy Heaster, Claire Collins, Canan Tuz Yilmaz, Felicity Knights, Maria de la Cruz Gomez Pellin, Maria Pilar Astier Peña, Neophytos Stylianou, Raquel Gomez Bravo, Venija Cerovečki, Zalika Klemenc Ketis, Sara Willems

**Affiliations:** 1Department of Public Health and Primary Care, Ghent University, 9000 Ghent, Belgium; 2Department of Family Medicine, School of Medicine, University of Zagreb, 10000 Zagreb, Croatia; 3Health Centre Zagreb-Centar, 10000 Zagreb, Croatia; 4Institute of General Practice, First Medical Faculty, Charles University, CZ-121 08 Prague, Czech Republic; 5Department of Family Medicine, Faculty of Medicine, Riga Stradiņš University, LV-1007 Riga, Latvia; 6Research Centre, Irish College of General Practitioners, D02 XR68 Dublin, Ireland; 7Family Medicine Department, Bursa Uludag University, 16130 Bursa, Turkey; 8Population Health Research Institute, St George’s University of London, London SW17 0RE, UK; 9Department of Social and Preventive Medicine, Medical University of Vienna, 1090 Vienna, Austria; 10Primary Health Centre Universitas, Aragon Health Services, 50009 Zaragoza, Spain; 11Medical School, Universidad de Zaragoza, GIBA-ISS-Aragón, 50009 Zaragoza, Spain; 12RTD Talos, 2404 Nicosia, Cyprus; 13International Institute for Compassionate Care, 2415 Nicosia, Cyprus; 14CHNP, Rehaklinik, L-9002 Ettelbruck, Luxembourg; 15Research Group Self-Regulation and Health, Institute for Health and Behaviour, Department of Behavioural and Cognitive Sciences, Faculty of Humanities, Education and Social Sciences, University of Luxembourg, L-4366 Esch-sur-Alzette, Luxembourg; 16Ljubljana Community Health Centre, 1000 Ljubljana, Slovenia; 17Department of Family Medicine, Medical Faculty, University of Maribor, 2000 Maribor, Slovenia; 18Department of Family Medicine, Medical Faculty, University of Ljubljana, 1000 Ljubljana, Slovenia

**Keywords:** safety culture, patient safety, quality of care, primary health care, COVID-19, medical education, vocational training, PRICOV-19, infectious disease, multi-country, general practice

## Abstract

The day-to-day work of primary care (PC) was substantially changed by the COVID-19 pandemic. Teaching practices needed to adapt both clinical work and teaching in a way that enabled the teaching process to continue, while maintaining safe and high-quality care. Our study aims to investigate the effect of being a training practice on a number of different outcomes related to the safety culture of PC practices. PRICOV-19 is a multi-country cross-sectional study that researches how PC practices were organized in 38 countries during the pandemic. Data was collected from November 2020 to December 2021. We categorized practices into training and non-training and selected outcomes relating to safety culture: safe practice management, community outreach, professional well-being and adherence to protocols. Mixed-effects regression models were built to analyze the effect of being a training practice for each of the outcomes, while controlling for relevant confounders. Of the participating practices, 2886 (56%) were non-training practices and 2272 (44%) were training practices. Being a training practice was significantly associated with a lower risk for adverse mental health events (OR: 0.83; CI: 0.70–0.99), a higher number of safety measures related to patient flow (Beta: 0.17; CI: 0.07–0.28), a higher number of safety incidents reported (RR: 1.12; CI: 1.06–1.19) and more protected time for meetings (Beta: 0.08; CI: 0.01–0.15). No significant associations were found for outreach initiatives, availability of triage information, use of a phone protocol or infection prevention measures and equipment availability. Training practices were found to have a stronger safety culture than non-training practices. These results have important policy implications, since involving more PC practices in education may be an effective way to improve quality and safety in general practice.

## 1. Introduction

The COVID-19 pandemic outbreak that was formally declared by the World Health Organization on 11 March 2020, had a profound impact on society and healthcare systems [1]. The pandemic disrupted routine clinical care, with many clinicians and trainees being redeployed away from their specialty areas to provide care at the COVID-19 frontline [2,3,4]. In addition to other changes, increased requirements for healthcare services due to COVID-19 challenged the balance between clinical practice and medical education for health professionals [2,3].

The pandemic had a significant impact on all levels of medical education, with the need for rapid adaptation to “blended learning”, using digital training interventions in addition to traditional teaching models [5,6]. Medical students and residents jointly participated in the fight against the pandemic in many places, supporting care provision to patients with COVID-19, but also acquiring important new competencies in fighting pandemics in the community [5].

Family medicine and other residency programs strived to develop strategies for keeping residents and patients safe, while maintaining high standards of education [4,7,8,9,10]. The rapid restructuring included development of risk-stratification guidelines, triage protocols, deferral of non-urgent patients and the transition of outpatient clinics to a telehealth model [4,7,8,9,10]. Teachers suddenly needed to adapt to new teaching formats, challenged with the burden of combining high-quality medical training with clinical work and the attempt to provide safe and high-quality patient care [11]. 

There is evidence that the reorganization of primary health care due to COVID-19 has led to reduced access to primary care (PC) practices, potentially resulting in unfavorable patient health outcomes, especially for vulnerable patient populations [12]. Some previous studies reported on the positive associations between practice training status and features of high-quality care in terms of practice organization, chronic care and preventive services [13,14]. Training practices in France showed better performance in diabetes follow-ups, seasonal flu vaccinations, and cervical cancer and breast cancer screenings [14]. In the Netherlands, training practices made better use of team skills, offered a wider range of diagnostic and therapeutic services and scored higher on disease management for diabetes and CVD than non-training practices [13]. Whether this positive effect of being a training practice on the quality of patient care is preserved in the time of pandemics is not known. Additionally, in the pre-COVID era, trainers reported higher levels of job satisfaction and commitment and lower levels of job stress in comparison to non-trainers [13]. As for stress levels during COVID-19, some studies reported high levels of stress in teachers with regards to providing medical training, but with a self-perception of adequate situational coping [11]. According to the World Health Organization (WHO), one of the ten key actions regarding improving safety in primary health care includes strengthening the workforce, as professional burnout, fatigue and stress can all adversely affect patient safety [15]. 

Patient safety in primary health care is a major concern in four main areas: diagnosis, prescribing, communication and organizational characteristics of primary care [16]. In a prospective French study prior to COVID-19, the incidence of reported patient safety incidents was 26 per 1000 patient encounters per week; the incidents were three times more frequently related to the organization of healthcare, especially to the workflow and communication between providers and patients, than to the knowledge and skills of health professionals [17]. The issues addressed by Flemish general practitioners (GPs) as potentially compromising quality of care and patient safety during COVID-19 included remote assessment and management of patient needs, changing focus of acute care to COVID-19 with patients consulting less frequently for non-COVID problems and postponing most care of chronic conditions [18]. The majority of safety incidents reported in French PC practices during COVID-19 were related to delayed diagnosis, assessments and referrals, but also to delayed consultation of primary healthcare providers due to patients’ fear of contracting the COVID-19 infection [19].

There are several theoretical frameworks for safety described in the literature, with significant overlaps in their understanding of what constitutes patient safety. Since our study focuses on the safety culture in primary health care, we drew from the Institute for Healthcare Improvement (IHI) Framework for Safe, Reliable, and Effective Care; the Manchester Patient Safety Framework (MaPSaF)–Primary Care Module; and WHO’s Technical Series on Safer Primary Care to delimit four dimensions of safety culture assessed in the PRICOV-19 study: “safe practice management”, “community outreach”, “professional well-being” and “adherence to protocols” [15,20,21]. The aim of this paper is to investigate the possible differences in the safety culture of training and non-training PC practices across countries.

## 2. Materials and Methods

### 2.1. Study Design and Setting

In the summer of 2020, an international consortium of more than 45 research institutes was formed under the coordination of the ‘Equity in Health care’ research unit at Ghent University (Belgium) to set up the PRICOV-19 study. This multi-country cross-sectional study aimed to research how primary care practices were organized during the COVID-19 pandemic to guarantee high-quality care; how the task roles changed and the pandemic impacted the wellbeing of care providers and whether differences could be found between types of practices and/or healthcare systems. Data were collected in 37 European countries and Israel. The described multi-country study design focusing on the organization, quality and safety of PC practices during COVID-19 provided an opportunity to assess whether practice training status was related to differences in safety culture outcomes in PC practices across countries. 

### 2.2. Measurement

Data was collected by means of an online self-reported questionnaire among PC practices. The questionnaire was developed at Ghent University in multiple phases, including a pilot study among 159 GP practices in Flanders (Belgium). More details are described elsewhere [22]. The questionnaire consists of 53 items divided into six topics: (a) infection prevention; (b) patient flow for COVID- and non-COVID care; (c) dealing with new knowledge and protocols; (d) communication with patients; (e) collaboration and wellbeing of the respondent; and (f) characteristics of the respondent and practice [22]. The questionnaire was translated into 38 languages following a standard procedure. The Research Electronic Data Capture (REDCap) platform was used to host the questionnaire in all languages, send out invitations to the national samples of GP practices and securely store the answers from the participants [23].

### 2.3. Sampling and Recruitment

Data was collected between November 2020 and December 2021, except for Belgium, where data was partially collected earlier. Data collection varied in duration between countries from 3 to 35 weeks. In each partner country, the consortium partner(s) recruited PC practices following a pre-defined recruitment procedure. Drawing a randomized sample among all PC practices in the country was preferred over convenience sampling. Partners logged all the steps taken in the sampling procedure. PRICOV-19 aimed to sample between 80 and 200 PC practices per country, depending on the national number of PC practices. However, since there was no funding for this study and coordinators recruited practices voluntarily, it was not possible to enforce a specific recruitment strategy or specific response rates. Per PC practice, one questionnaire was completed, preferably by a GP or by a staff member familiar with the practice organization. The overall response rate was 27.8% ranging from 1.55% in Denmark to 94.3% in Bulgaria [24].

### 2.4. Definition of Training Practice

Training practices were identified in the sample based on the question “How many GPs and GP trainees are working in this practice?”. If the respondent indicated having one or more GP trainees working in their PC practice, the practice was flagged as a training practice.

### 2.5. Description of Outcomes Studied

Outcome variables were selected based on their relevance in relation to four dimensions of safety culture in PC practices. The dimensions were “safe practice management”, “community outreach”, “professional well-being” and “adherence to protocols”. For “safe practice management”, the outcomes identified were: total amount of infection prevention equipment available, total number of patient flow safety measures since COVID-19, total number of infection prevention measures since COVID-19 and total number of different safety incidents that occurred since COVID-19. For the dimension “community outreach”, the outcome identified was the total number of different outreach initiatives taken since the start of the pandemic. For the dimension “professional well-being”, the outcome identified was the Mayo Clinic Well-being Index (MCWI), calculated and classified as “at high risk for adverse mental health events” vs. “not at high risk” based on the MCWI manual [25]. For the dimension “adherence to protocols”, the outcomes identified included availability of information on triage centers at the GP’s office, use of a telephone protocol when assessing possible COVID-19 patients by phone and how often there was protected time for meetings to review guidelines during COVID-19. The full description of the questions used and their defining statements can be found in Appendix A.

### 2.6. Variable Coding

Variables of interest were selected based on the literature and used as control variables. These included the following variables: respondent’s years of experience coded in ten year groupings; being a multidisciplinary practice, questioned as “yes or no” (referring to practices that had at least one health professional of another discipline besides medicine and nursing); location coded as “big (inner) city”, “suburbs or (small) town” and “mixed urban-rural or rural”; number of GPs coded as “solo”, “dual” and “group” practices; and function of the respondent coded as “GP” and “GP trainee” (only included in the regression for MCWI).

### 2.7. Data Analysis

Categorical variables were described according to their frequency in the sample, whereas numerical variables were described based on their mean and standard deviation (SD). Bivariate analyses were performed to test for the association between being a training practice or not, and the outcomes were studied according to variable type. A Pearson’s chi-square (χ^2^) test was computed for binary and count variables (infection prevention equipment, safety incidents, outreach initiatives, MCWI score, triage information, phone protocol) and an independent two-sided t-test for numerical variables (patient flow safety measures, infection prevention measures, protected time for meetings). To investigate the effect of being a training practice on selected safety-related outcomes, we built mixed-effects regression models using the variable “country” as a random intercept, in order to account for the different distribution of training practices across countries and the clustered nature of the data. For each outcome variable previously described, a regression model was built according to the variable’s type and distribution (see Table 1).

Descriptive and bivariate statistical analyses were performed using SPSS software (version 28.0 SPSS Inc., Chicago, IL, USA). The mixed-effects regression analysis was made using the following software: R Core Team, R software version 4.0.3 (R Foundation for Statistical Computing, Vienna, Austria); RStudio version 1.3.1093 (PBC, Boston, MA, USA). Ghent University was responsible for the data cleaning of the international data and database version 7 was used for this analysis, consisting of the cleaned data of 33 countries.

### 2.8. Ethics Approval

The study was conducted according to the guidelines of the Declaration of Helsinki. The Research Ethics Committee of Ghent University Hospital approved the protocol of the PRICOV-19 study and Belgian data collection (BC-07617). Research Ethics Committees in the different partner countries gave additional approval if needed. All participants gave informed consent on the first page of the online questionnaire.

## 3. Results

At the time of this analysis, a total of 8,161 GP practices from 32 European countries and Israel had answered the PRICOV-19 questionnaire. Two thirds (5158/8161) provided an answer to the question on the number of GP trainees working in their practice. Among them, 2272 (44.0%) were defined as training practices and 2,886 (50.6%) were not. The proportion of training practices varied widely across countries, from 13.9% in Moldova to 94.2% in Sweden. Distribution of training and non-training practices per country can be found in Table 1. 

Table 2 shows the percentages for the binary and count variables or the mean (standard deviation) for the numerical of training and non-training practices among the different outcomes and the significance of the chi-square or t-test. Significant associations were found for risk for adverse mental health events according to the MCWI score, number of outreach initiatives, number of patient flow safety measures in place since COVID-19 and number of safety incidents (all *p*-values < 0.01). Table 3 shows the distribution of the selected covariates (work experience in primary health care, multidisciplinarity of the practice, practice location, number of GPs in the practice and function of the respondent) among training and non-training practices in the sample and the associated *p*-value.

### Mixed-Effects Regression Models

Table 4 shows the results of the mixed-effects regression analyses, including “Country” as a random intercept. The coefficients represent the effect of being a training practice in each of the outcomes, with its respective *p*-value, coefficient and 95% confidence interval. All models were controlled for the following variables: respondent’s years of experience, being a multidisciplinary practice, urban or rural location and number of GPs. The model for being at risk for adverse mental health events had an additional covariate controlling for function of the respondent (GP or GP trainee).

Being a training practice had a protective effect on adverse events according to the MCWI score (OR: 0.83, 95% CI: 0.70–0.99). A positive effect of being a training practice was also found in the number of safety measures for preventing infection transmission among patients and staff, with a 0.17 increase in the number of safety measures among training practices when compared to non-training practices (95% CI: 0.07–0.28). The number of reported safety incidents was higher in training practices when compared to non-training practices (RR: 1.12, 95% CI: 1.06–1.19), as was the frequency of protected time for meetings (Beta: 0.08, 95% CI: 0.01–0.15). Outreach initiatives undertaken by practices since the start of the COVID-19 pandemic were more common among training practices (RR: 1.06, 95% CI: 0.99–1.12), however, this difference did not achieve significance. Similarly, no significant effect was found for training practices on the amount of infection prevention equipment, availability of triage information, the use of a COVID-19 phone protocol and infection prevention measures during COVID-19 (*p*-values respectively: 0.83; 0.78; 0.51; 0.15).

## 4. Discussion

This is the first cross-country analysis on the safety culture of PC practices during the COVID-19 pandemic with regards to the teaching status of practices. The results support the hypothesis that training practices had a more favorable safety culture than non-training practices during the pandemic. We found that training practices had a significant association with not being at high risk of adverse mental health outcomes for the caregiver himself/herself, a higher number of patient flow safety measures to lower the risk of infection transmission in the practice and higher reporting of safety incidents. Training practices also had significantly more protected time for meetings to review guidelines and reflect on practice management. These results were confirmed after controlling for respondent’s years of experience, being a multidisciplinary practice, urban or rural location and number of GPs in the practice. Alternatively, no significant associations were found with the amount of infection prevention equipment, general infection prevention measures, number of outreach initiatives, the availability of triage information or the use of a COVID-19 phone protocol.

### 4.1. Safe Practice Management

Infection prevention and control measures gained the spotlight among safety measures during COVID-19 [26]. Our study investigated such measures under two different lenses: general infection prevention measures during COVID-19 (such as staff wearing accessories or availability of hand sanitizer) and reorganization of patient flow in the practice (such as waiting room and reception redesign or use of online tools to reduce in-person contact). No significant difference was found for the former. This can be attributed to the fact that measures evaluated in this outcome were standard measures and already part of infection control routines in primary health care before the pandemic. Additionally, no difference was found on the presence of infection prevention equipment between training and non-training practices. Previous studies on the safety of primary health care in Europe showed that the presence of such equipment was already high among most European PC practices [27].

On the other hand, the number of patient flow safety measures was significantly higher in training practices when compared to non-training practices. The measures evaluated in this outcome have a closer relationship with the context of the coronavirus pandemic, thus not being standard practices previously. The fact that training practices adhered to such measures more often than non-training practices indicates they are more finely attuned to good practice recommendations and the adoption of innovative (soft) technologies [14,28]. Training practices are often larger and have more staff than non-training practices [29,30], and they may have more time to review guidelines: both aspects are supported by our findings, and this may lead to a higher uptake of such new measures. However, even after controlling for having a multidisciplinary team or a larger practice, the effect of being a training practice persisted, which points to an association with the educational profile of the practice itself and the implementation of safety measures.

Training practices reported more safety incidents according to the PRICOV-19 questionnaire. This finding may be interpreted in two opposite ways: either as evidence of a less safe practice or as an indication of a better safety culture, with a lower threshold to report and communicate openly about safety incidents. GP trainees progress through different stages of learning over the course of training, and this may lead to more frequent unsafe practices, which should be identified and rectified; higher reporting of safety incidents in the training practices may therefore indicate the presence of a higher overall number of safety incidents and a less safe practice. However, having a culture of self-reflection is an important part of safety in primary health care. Being able to identify safety incidents is intrinsic to the process of quality and safety improvement and such transparency ultimately leads to increased safety in delivering care [21]. The trainer–trainee relationship entails having dedicated time to review patients and reflect on the trainee’s practice [31], which can also lead to higher identification and reporting of safety incidents. 

### 4.2. Community Outreach

Although outreach initiatives were more common among training practices, with a significant association in the initial bivariate analysis, this association did not achieve significance in the regression analysis (RR = 1.06 *p* = 0.07). During COVID-19, such initiatives were of heightened importance, given the reduction in physical contact especially in the initial stages of the pandemic [12]. Many studies reported on loss of continuity of care due to social distancing measures or to fear of contamination [12,32,33,34]. In this context, outreach initiatives gained attention as a safety practice. It is possible that unidentified confounders also played a role in the uptake of outreach initiatives by individual primary care practices, in particular the degree of vulnerability of its patient population.

### 4.3. Professional Well-Being

Although studies showed the mental health of students, trainees and faculty have suffered from the impact of COVID-19 [2,4,35,36], our results show that working in a training practice lowered the risk of adverse mental health events. This effect persists even after controlling for having a multidisciplinary team, the respondent’s years of experience in primary health care, number of GPs in the practice, urban or rural location and function of the respondent in the practice (GP or GP trainee). This finding might have relevant implications in terms of the relationship between professional well-being and patient safety outcomes, as advocated by the WHO [15]. Although research in this area is scarce and findings diverse, several studies found an association between physician burnout, a negative indicator of professional well-being, and an increased likelihood of perceived errors [37,38,39]. As opposed to that, recent research by Lu et al. has shown how institutional patient safety culture leads to better staff well-being by reducing burnout and enhancing work–life balance [40]. In the literature, better professional well-being is associated with good relationships inside the practice and strong teamwork [41,42,43]. Doctors working in training practices may experience more professional support and less isolation, which is related to increased resilience [44]. There may also be an increase in job satisfaction related to the teaching aspect of their job [45]. Alternatively, GPs who choose to be trainers may already have higher resilience and intrinsic motivation, although little evidence of this exists in the published literature [45].

### 4.4. Adherence to Protocols

Our data shows no significant differences for the availability of COVID triage information in the GP’s office or use of a phone protocol for patients suspected of being infected with COVID-19. These two practices are likely to have been widely adopted across primary care practices, independent of being a training practice or not [26]. Nonetheless, when analyzing the effect of being a training practice on the frequency of protected time for meetings to review guidelines, a significant association was found. Having dedicated time for meetings related to practice organization, patient management and reviewing guidelines is an important characteristic of training practices [31] and one that is closely related to continuous education and self-reflection. This particular trait may be key to enhancing professional well-being, implementing safety measures and understanding the significant differences in the other dimensions of safety previously mentioned [46].

### 4.5. Study Limitations

This study presents with several limitations. Firstly, PRICOV-19 relied on volunteer participation, which can lead to self-selection in favor of practices with an interest in quality and safety of care, and possible overrepresentation of teaching practices in the sample of some countries (the proportion of GP training practices at the national level according to the literature varies between 10% to 30% in the EU) [14,47,48], possibly because universities were often the main recruiters. As PRICOV-19 was a non-funded study, not all partner countries could recruit a randomized sample to participate. Furthermore, data was collected through an online questionnaire, with no direct observation of practice organization and functioning, relying on the self-reporting of respondents, which can also lead to recall bias. Moreover, conclusions about causal relationships were hampered by the cross-sectional design of the study. The regression models in the data analysis controlled for significant covariates, including work experience in primary health care; however, mean duration of experience for each practice was represented by the individual completing the survey when years of experience for the entire practice would have been a more accurate representation. In addition, the timeframe of data collection was rather wide, reflecting different stages of the developing pandemic, including changes in social distancing measures, the roll-out of vaccination programs and changes in well-being of practitioners. However, the available database did not allow for accurately establishing the precise burden of COVID-19 in each country when data was collected.

## 5. Conclusions

Our data has established a broad positive association between being a training practice and several key safety-related outcomes. This means training young GPs has an important positive impact on the health system. It safeguards the health workforce of the future (and the present), while also being associated with higher quality and safety of the practices involved in training while lowering the risk of distress for qualified GPs participating in vocational training. Future qualitative research could investigate in depth why training practices are doing better in certain outcomes and what underlying mechanisms are involved. Understanding the principles of adaptability of training practices to lower the risk of infection transmission could be important in PC preparations for future pandemics. Such benefits also have important policy implications: getting more PC practices involved in becoming training practices might be an effective way to increase quality and safety in primary health care. Participation of PC practices in vocational training might also be an effective intervention to reduce GPs’ risk of adverse mental health outcomes and ensure the sustainability of the primary care workforce which is essential for both the functioning of the health care system, and society as a whole.

## Figures and Tables

**Table 1 ijerph-19-10515-t001:** Distribution of training practices across countries, total and valid percentage.

Country	Non-Training Practice	Training Practice	Total
Austria	109 (78.4%)	30 (21.6%)	139
Belgium	278 (58.0%)	201 (42.0%)	479
Bosnia and Herzegovina	24 (60.0%)	16 (40.0%)	40
Bulgaria	71 (69.6%)	31 (30.4%)	102
Croatia	112 (75.2%)	37 (24.8%)	149
Czechia	76 (69.1%)	34 (30.9%)	110
Denmark	11 (28.2%)	28 (71.8%)	39
Estonia	84 (71.2%)	34 (28.8%)	118
Finland	13 (11.3%)	102 (88.7%)	115
France	335 (52.3%)	306 (47.7%)	641
Germany	145 (55.3%)	117 (44.7%)	262
Greece	61 (65.6%)	32 (34.4%)	93
Hungary	181 (80.8%)	43 (19.2%)	224
Iceland	9 (29.0%)	22 (71.0%)	31
Ireland	107 (57.8%)	78 (42.2%)	185
Israel	36 (40.0%)	54 (60.0%)	90
Italy	114 (55.3%)	92 (44.7%)	206
Kosovo *	16 (20.5%)	62 (79.5%)	78
Latvia	125 (84.5%)	23 (15.5%)	148
Lithuania	42 (82.4%)	9 (17.6%)	51
Malta	7 (53.8%)	6 (46.2%)	13
Moldova	62 (86.1%)	10 (13.9%)	72
the Netherlands	95 (57.2%)	71 (42.8%)	166
Norway	88 (61.1%)	56 (38.9%)	144
Poland	80 (37.9%)	131 (62.1%)	211
Portugal	49 (21.8%)	176 (78.2%	225
Romania	79 (78.2%)	22 (21.8%)	101
Serbia	51 (44.7%)	63 (55.3%)	114
Slovenia	126 (65.3%)	67 (34.7%)	193
Spain	117 (38.9%)	184 (61.1%)	301
Sweden	5 (5.8%)	81 (94.2%)	86
Switzerland	71 (81.6%)	16 (18.4%)	87
Turkey	107 (73.8%)	38 (26.2%)	145
Missing cases	-	-	3003
Total	2886 (56.0%)	2272 (44.0%)	8161

* All references to Kosovo, whether the territory, institutions or population, in this project, shall be understood in full compliance with United Nations Security Council Resolution 1244 and the ICJ Opinion on the Kosovo declaration of independence, without prejudice to the status of Kosovo.

**Table 2 ijerph-19-10515-t002:** Distribution of outcome variables among training and non-training practices and bivariate analysis.

Outcome	Non-Training Practice	Training Practice	*p*-Value
**Number of infection prevention equipment in the practice**			
0	0 (0%)	1 (0%)	0.15
1	1 (0%)	1 (0%)	
2	8 (0.3%)	14 (0.3%)	
3	51 (2%)	48 (2.3%)	
4	147 (5.7%)	130 (6.3%)	
5	535 (20.9%)	465 (22.7%)	
6	944 (36.8%)	729 (35.6%)	
7	877 (34.2%)	662 (32.3%)	
**Number of patient flow safety measures in place since COVID-19**			
Mean	4.5403	4.8142	<0.01
Standard deviation	1.7612	1.7384	
Total cases	2506 (55.4%)	2013 (44.6%)	
**Infection prevention measures**			
Mean	4.1988	4.1396	0.15
Standard deviation	1.3595	1.3796	
Total cases	2561 (55.6%)	2048 (44.4%)	
**Safety incidents**			
0	908 (36.1%)	600 (29.9%)	<0.01
1	628 (25.0%)	479 (23.9%)	
2	477 (19.0%)	417 (20.8%)	
3	272 (10.8%)	264 (13.2%)	
4	141 (5.6%)	142 (7.1%)	
5	89 (3.5%)	105 (5.2%)	
**Number of outreach initiatives**			
0	941 (36.3%)	634 (30.9%)	<0.01
1	629 (24.3%)	482 (23.5%)	
2	592 (22.8%)	516 (25.2%)	
3	323 (12.5%)	302 (14.7%)	
4	108 (4.2%)	117 (5.7%)	
**Risk for adverse mental health events according to the Mayo Clinic Well-Being score**			
Not at high risk for adverse outcomes	694 (30.4%)	641 (35.0%)	<0.01
At high risk for adverse outcomes	1588 (69.6%)	1188 (65.0%)	
**Is information on COVID triage protocol and centers available for GPs in their consultation rooms?**			
No	500 (22.7%)	410 (23.2%)	0.70
Yes	1707 (77.3%)	1360 (76.8%)	
**Does this practice use a protocol when answering the phone?**			
No	659 (25.7%)	475 (23.5%)	0.10
Yes	1907 (74.3%)	1542 (76.5%)	
**Protected time for meetings**			
Mean	1.6797	1.7240	0.16
Standard deviation	1.0697	0.9823	
Total cases	2432 (55%)	1993 (45%)	

GPs = general practitioners.

**Table 3 ijerph-19-10515-t003:** Distribution of covariates among training and non-training practices.

	Non-Training Practice	Training Practice	*p*-Value
**Work experience in primary health care-categorical-groups of 10 years (*n* = 4681)**			
0 to 9 years 11 months	781 (26.4%)	527 (26.9%)	0.01
10 years to 19 years 11 months	629 (23.1%)	527 (26.9%)	
20 years to 29 years 11 months	774 (28.4%)	506 (25.8%)	
30 years or more	602 (22.1%)	398 (20.3%)	
**Is this practice multidisciplinary? (*n* = 5097)**			
Monodisciplinary	2133 (75.1%)	1189 (52.7%)	<0.001
Multidisciplinary	707 (24.9%)	1068 (47.3%)	
**How would you characterize the place of this practice? (*n* = 5119)**			
Big (inner)city	913 (32.0%)	765 (33.8%)	0.002
Suburbs or (Small) town	789 (27.6%)	691 (30.5%)	
Mixed urban-rural or Rural	1154 (40.4%)	807 (35.7%)	
**How many GPs are working in the practice? (*n* = 4922)**			
Solo	1379 (48.6%)	356 (17.1%)	<0.001
Duo	474 (16.7%)	300 (14.4%)	
Group	984 (34.7%)	1429 (68.5%)	
**What is your position in this practice? (*n*= 4708)**			
GP	2663 (99.8%)	1739 (85.3%)	<0.001
GP trainee	6 (0.2%)	300 (14.7%)	

GPs = general practitioners.

**Table 4 ijerph-19-10515-t004:** Effect of being a training practice on selected outcomes.

Outcome	*p*-Value	Coefficient (CI) ^3^
Risk for adverse mental health events ^1^	0.04	OR: 0.83 (0.70–0.99)
Total sum of infection prevention equipment ^2^	0.83	RR: 1.00 (0.97–1.03)
Total sum of safety measures in place ^2^	<0.01	Beta: 0.17 (0.07–0.28)
Total sum of outreach initiatives ^2^	0.07	RR: 1.06 (0.99–1.12)
Availability of triage information ^2^	0.78	OR: 0.97 (0.81–1.17)
Use of a phone protocol ^2^	0.51	OR: 1.06 (0.89–1.25)
Infection prevention measures ^2^	0.15	Beta: −0.07 (−0.16–0.02)
Safety incidents ^2^	0.01	RR: 1.12 (1.06–1.19)
Protected time for meetings ^2^	0.02	Beta: 0.08 (0.01–0.15)

^1^ Controlled for respondent’s years of experience, being a multidisciplinary practice, urban or rural location, number of GPs in the practice and function of the respondent. ^2^ Controlled for respondent’s years of experience, being a multidisciplinary practice, urban or rural location and number of GPs in the practice. **^3^** OR: Odds Ratio (logistic regression); RR: Relative Risk (Poisson regression); Beta: Linear Regression Coefficient.

## Data Availability

The anonymized data is held at Ghent University and is available to participating partners for further analysis upon signing an appropriate usage agreement.

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
