# Peer review of "Safety Culture and the Positive Association of Being a Primary Care Training Practice during COVID-19: The Results of the Multi-Country European PRICOV-19 Study"

_ijerph, 2022, doi:10.3390/ijerph191710515_

Round 1
Reviewer 1 Report
This is a novel study focusing on the impact of practice training status on safety outcomes.
Overall – Minor additional English language review should be conducted.
Introduction:
Page 1, line 38 – “Has had a profound impact”, not just in past tense. We are still experiencing the COVID pandemic.
Page 2, line 48 – Should “competences” be “competencies?”
What percent of family practice sites provide educational opportunities for medical students? Is this a typical practice offering?
Page 2, line 59 – What positive associations have been noted in research focusing on the association between practice training status and practice organization, chronic care, and preventive services?
Authors need to provide a clearer connection between trainer stress and safety issues. They touch on trainers reporting higher job satisfaction prior to the pandemic, but provide no connection to patient safety outcomes.
Authors need to provide additional information on patient safety prior to and during the pandemic – for both training practices and non-training practices.
Materials and Methods:
Page 2, line 91 – Use the entirety of the term general practitioner first prior to using its abbreviation.
As only one individual completed the survey and represented the entire practice (training and non-training), how would number of years of experience impact patient safety? Authors should provide a justification for controlling for this variable. It would appear that providing the mean number of years of experience for the entire practice would be a better measure?
Results:
Page 4, line 172 – The wording of the statement is confusing – “2/3 stated to have or not GP trainings in their practice.” Does this mean that 2/3 had GP trainees? Or that 2/3 did not have GP trainees?
Table 3 - Authors should provide stratification of covariates by training/non-training status, not just overall.
Discussion:
Authors need to provide additional explanation of how higher reporting of safety incidents is a positive outcome for training practices. Is this the result of a higher reporting of safety incidents, with the same number of overall safety incidents for both training and non-training practices, or is this indicative of a higher overall number of safety incidents?
Page 8, line 254 – Authors mention that training practices are often larger and have more staff, and as a result they many have more time to review guidelines. Within this study, were the training practices larger?
Page 8, lines 260-270 – This is an important paragraph, and is the basis for many conclusions drawn within this paper. Authors need to see this as a limitation of the study. The report of more safety incidents is not necessarily the result of being a more safe practice, and this needs to be emphasized.
Professional well-being – Authors need to draw a connection between professional well-being and safety (throughout paper), otherwise, the results related to this should be excluded from the paper, as there appears to be no connection at this point to patient safety.
Reviewer 2 Report
This is an interesting paper, which studies novel area. Another strong point about the paper is wide and sound/representative sample of the study. Given above, the paper is valuable and should be considered to be published. Before that, however, there are several major technical issues to be adjusted. Following their implementation the paper shall get ready to be published.
My detailed comments are as follows
1. Please improve the language in the whole paper
2. Please delete in abstract "(1) Background", "(2) Methods" and so on
3. In the very beginning of Section 2 (Materials and methods) please explain why this approach is the most suitable to achieve the study objective.
4. Please attache the questionnaire, which consists of 53 items divided into six topics - as appendix
5. Please provide more implications of the study to various stakeholders in the conclusions section
6. Please improve the layout of Appendix A1
Round 2
Reviewer 2 Report
The authors have done a good job implementing requested adjustments. The paper can now be published.